# Elder Mistreatment as a Risk Factor for Depression and Suicidal Ideation in Korean Older Adults

**DOI:** 10.3390/ijerph191811165

**Published:** 2022-09-06

**Authors:** Min-So Paek, Mi Jin Lee, Yu-Seon Shin

**Affiliations:** Department of Social Welfare, Konkuk University, Chungju-si 27478, Korea

**Keywords:** elder mistreatment, elder abuse, multi-type mistreatment, depression, suicidal ideation

## Abstract

Older adults suffering from mistreatment are especially vulnerable to adverse health outcomes. The current study examined the associations of elder mistreatment (single- and multi-type mistreatment) with depression or suicidal ideation in a Korean representative sample. The data were derived from the 2017 National Survey of Living Conditions and Welfare Needs of Korean Older Persons (unweighted *n* = 10,059 and weighted *n* = 10,055). Descriptive statistics and multivariate logistic regression analyses were performed. In the weighted population, 9.8% of older adults had mistreatment experiences. Results indicated that single- and multi-type mistreatment experiences were associated with increased risks of depression (OR = 1.93, 95% CI = [1.61, 2.32] and OR = 3.51, 95% CI = [2.52, 4.87], respectively), after adjusting for the confounding factors (socio-demographic, health-related, and social relation characteristics). Experiences of single- and multi-type mistreatment were also associated with suicidal ideation (OR = 2.48, 95% CI = [1.97, 3.12] and OR = 3.19, 95% CI = [2.25, 4.51], respectively), even after adjusting for the above confounding factors and depression. Similar results were found in sensitivity analyses using unweighted data. The current findings expanded our knowledge of the associations of mistreatment with depression and suicidal ideation in later life.

## 1. Introduction

Elder mistreatment is becoming a global social problem as the number and proportion of older adults continue to increase. About one in every six older adults is estimated to suffer from some type of mistreatment [1]. Elder mistreatment is defined as “an intentional act or failure to act by a caregiver or another person in a relationship involving an expectation of trust that causes or creates a risk of harm to an older adult” [2] and includes various types of mistreatment against older adults, such as physical, psychological, sexual, and financial mistreatment, neglect, abandonment, and any combination of the preceding.

In South Korea, elder mistreatment has long been recognized as a personal or family problem and has emerged as a social problem relatively late [3,4]. In 2004, the Welfare of Senior Citizens Act was amended to prevent and solve problems of elder abuse, and many Elder Protective Services agencies (EPSs) were established. Since that establishment, unexpectedly many cases of elder mistreatment were reported to the EPSs. The number of confirmed elder mistreatment cases increased by 204% from 3068 in 2010 to 6259 in 2020 [5,6]. Approximately 75% of the cases in 2020 reported experiencing two or more types of mistreatment (denoted “multi-type mistreatment”). With the considerable increase in the number and proportion of the older population in South Korea [7], the number of older adults who suffer from elder mistreatment is expected to continue to increase.

Elder mistreatment is associated with a wide range of physical and psychological health outcomes and well-being. Several studies have shown that mistreated older adults have a greater mortality risk [8], more frequent hospitalization and emergency department utilization [9,10], and worse physical health conditions (e.g., chronic pain, bone or joint problems) than non-mistreated counterparts [11,12]. Researchers pointed out that elder mistreatment could cause psychological consequences, which could be even more devastating than physical health sequelae [13,14]. Findings from systematic review studies suggested that poor mental health was a common psychological problem observed among mistreated older adults and they were at higher risk than their non-mistreated counterparts for developing depression, anxiety, and posttraumatic stress disorder [13,15]. Yet not enough is known about the association between elder mistreatment and suicidal ideation, and a few studies suggested that elder mistreatment might be associated with an increased risk of suicidal ideation [16,17,18].

Most of this substantial knowledge mainly focuses on single types of elder mistreatment. While there is some evidence that exposures to multi-type elder mistreatment may be associated with an even higher increased risk of adverse mental health consequences, the area remains substantially understudied [12,19,20]. Furthermore, there is very limited research that is focusing on the impact of single- and multi-type mistreatment on mental health in the Korean older population [21].

### Study Objectives

Given that mistreated older adults in South Korea are a high-risk and understudied group, further research is needed. Using a nationally representative sample of Korean older adults, the current study aims to examine the associations of elder mistreatment with depression or suicidal ideation. We hypothesize that (1) older adults with mistreatment experiences (single- and multi-type mistreatment) have a higher increased risk of depression and suicidal ideation than those without mistreatment experiences and (2) multi-type elder mistreatment would be greatly associated with increased risks of depression and suicidal ideation.

## 2. Methods

### 2.1. Data Sources and Study Population

Data were derived from the “2017 National Survey of Living Conditions and Welfare Needs of Korean Older Persons” conducted by the Korean Ministry of Health and Welfare and the Korea Institute for Health and Social Affairs (KIHASA). The survey is a large national probability sample representing the current living conditions, health, and social status of older adults aged 65 years and over. The data contain information on various life domains such as family structure, physical and mental health, financial situation, and social life. The details of the methodology including sampling procedures are described by the KIHASA [22]. A two-stage stratified cluster sampling method was used to select a sample of approximately 10,000 older adults from 17 provinces in South Korea. As described in the manual of KIHASA, a sampling weight is assigned to each person in the survey for generalizability to the older population [22]. The survey has been conducted every three years since 2008, and the data collected in 2017 included 10,299 older individuals. Of them, 216 individuals who did not directly participate in the survey were excluded because there were no data on elder mistreatment, and 24 respondents who have missing data on the study variables were excluded from the current study. The final data included 10,059 self-respondents. The current study was exempted by Konkuk University IRB as this was a secondary analysis of deidentified public data.

### 2.2. Study Measures

Elder mistreatment. Elder mistreatment was assessed using six questions developed for the survey, which measured the occurrence of physical, psychological, sexual, and financial mistreatment and two types of neglect (e.g., neglect in primary needs including financial neglect and care neglect). Respondents answered either “yes (coded as 1)” or “no (coded as 0)” to the following questions: During the past 12 months, (1) “Have you ever experienced physical mistreatment such as being beaten and pushed?”, (2) “Have you ever experienced psychological mistreatment such as avoidance of conversations, ignoring opinions, and pretending not to hear?”, (3) “Have you ever experienced sexual mistreatment such as sexual violence?”, (4) “Have you ever experienced financial mistreatment such as spending money without permission?”, (5) “Didn’t your family members or caregivers give you any living expenses?”, and (6) “Didn’t your family members or caregivers provide care even when you were sick?”. The scores were summed (range = 0–6) and respondents were classified by the number of mistreatment categories they experienced: no mistreatment (coded as 0), single-type mistreatment (coded as 1), and multi-type mistreatment (≥2) groups.

Depression. Depression was assessed using the Korean version of the 15-item short form of the Geriatric Depression Scale (K-SGDS), a widely used tool to screen for depression in community-dwelling older adults [23]. Each item (e.g., “During the last week, do you feel that your situation is hopeless?”) was responded to dichotomously (no = 0, yes = 1) and the total score ranges from 0 to 15. Five positively phrased items were reverse-scored before summing. Respondents were divided into two groups, depressed (≥8; coded as 1) and non-depressed (<8; coded as 0), as 8 is the recommended optimal cut-off value for the K-SGDS [23].

Suicidal ideation. Suicidal ideation refers to thinking about, considering, or planning of ending one’s own life [24]. In this study, suicidal ideation was examined using a question: “Have you ever thought about suicide after the age of 60 years old?” The response categories were yes and no. Respondents who answered “yes” to the question were designated as having suicidal ideation and were coded as 1; those who answered “no” were classified as not having suicidal ideation and were coded as 0.

Potential confounding variables. The questionnaire included questions on socio-demographic and health-related characteristics, and the following risk factors which were associated with mental health outcomes (e.g., depression, and suicidal ideation) were taken into account: age (years), gender (male, female), education (below high school, high school and above), marital status (married, unmarried), living with children (yes, no), area (urban, rural), income (quintiles of annual household income; 1 = the lowest 20% and 5 = the highest 20%), doctor-diagnosed chronic disease (e.g., hypertension, diabetes, stroke) (yes, no), activities of daily living (ADL) and instrumental ADL (IADL) disabilities (yes, no), Mini-Mental State Examination (MMSE)-based cognitive impairment (yes, no; using different cut-off scores depending on age, gender, and education level) [22], self-rated health (poor, good), physical inactivity (yes, no), and drinking (yes, no). The following variables relating to social relations were also taken into account: social participation (participation in paid and volunteer work, hobby groups, and political organizations) (yes, no), social network (number of close siblings/relatives/friends/neighbors), and average frequency of contact with siblings/relatives/friends/neighbors (0 = never, 1 = 1–2 times per year, 2 = 1–2 times per 3 months, 3 = 1–2 times per month, 4 = once a week, 5 = 2–3 times per week, 6 = 4 days or more per week).

### 2.3. Statistical Analysis

Descriptive statistics (frequency distributions, percentage, mean, and standard deviation) were performed to describe the study sample as a whole and by key variables (e.g., single- and multi-type elder mistreatment, depression, and suicidal ideation). Two separate multivariate logistic regression analyses were conducted using hierarchical models to examine the associations of single-/multi-type mistreatment experiences with depression or suicidal ideation. In the multivariate logistic regression model, we simultaneously adjusted for potential confounding variables as covariates to control for possible effects of confounders. The following confounding variables were included: age, gender, education, marital status, living with children, area, household income, chronic disease, ADL/IADL disabilities, cognitive impairment, self-rated health, physical inactivity, drinking, social participation, social network, and frequency of contact. In the multivariate model for suicidal ideation, depression was further adjusted as a covariate in model 2 in addition to other potential confounding variables, because of its association with the increased risk for suicidal ideation. The Hosmer–Lemeshow goodness-of-fit test was used to determine the fit of the study model. Descriptive statistics and multivariate logistic regression analyses were performed using weighted data. A sensitivity analysis using unweighted data was also performed to check whether similar results can be obtained. A *p*-value of <0.05 was considered statistically significant. All statistical analyses were conducted using SPSS Statistics 27 (IBM, Armonk, NY, USA).

## 3. Results

### 3.1. Study Sample Characteristics

This study sample consisted of 10,055 (unweighted *n* = 10,059) older individuals. Table 1 describes the socio-demographic, health-related, and social relation characteristics. The mean age of the total sample was 73.87 years (*SD* = 6.54 years). Females were 57.5% and males were 42.5% of the entire sample. Most had less than a high school education (75.0%) and over half were married (63.7%). The majority did not live with their children (76.1%) and lived in urban areas (68.7%). Most had at least one chronic disease (89.5%), had no ADL/IADL disability (76.4%), and had no cognitive impairment (85.7%). More than half had poor self-rated health (63.0%) and 68.0% had physical inactivity. Over half of the respondents did not drink at all during the past 12 months (73.3%) and had social participation (60.1%). The mean social network size was 2.27 (*SD* = 2.65) and the mean frequency of contact with siblings/relatives/friends/neighbors was 3.05 (*SD* = 1.07).

In this study, the weighted prevalence of elder mistreatment was approximately 9.8% overall, 7.8% for single-type, and 2.0% for multi-type mistreatment. The weighted prevalence rate was 7.4% for psychological mistreatment, 2.3% for neglect in primary needs, 1.7% for care neglect, 0.4% for financial mistreatment, 0.3% for physical mistreatment, and 0.05% for sexual mistreatment. The weighted prevalence rates for depression and suicidal ideation were 21.0% and 6.7%, respectively. Using unweighted data, 9.8% reported experiencing elder mistreatment (7.9% for single- and 1.9% for multi-type), 21.0% had depression, and 6.7% had suicidal ideation.

### 3.2. Multivariate Logistic Regression Analyses

Table 2 presents the results of multivariate logistic regression analysis for depression. The multivariate logistic regression model controlled for potential confounding variables. After adjusting for covariates, older adults who experienced single-type mistreatment (odds ratio (OR) = 1.93, 95% confidence interval (CI) = [1.61, 2.32]) and those who experienced multi-type mistreatment (OR = 3.51, 95% CI = [2.52, 4.87]) had significantly higher odds for depression compared to the non-mistreated counterpart group (reference group). Most covariates included in the multivariate model were significantly associated with depression except for gender, marital status, cognitive impairment, and drinking. The Hosmer–Lemeshow goodness-of-fit statistic was not significant (*χ*^2^ = 8.82, df = 8, *p* = 0.36), indicating a good model fit.

Table 3 shows the results of multivariate logistic regression analysis for suicidal ideation. In model 1 after adjusting for covariates, older adults who experienced single-type mistreatment had 2.85 times (95% CI = [2.28, 3.56]) and those who experienced multi-type mistreatment had 4.21 times (95% CI = [3.01, 5.89]) greater odds for suicidal ideation compared to older adults without mistreatment experiences. The Hosmer–Lemeshow goodness-of-fit statistic was 4.36 (*p* = 0.82), indicating a good fit for the model. In model 2, the ORs were attenuated by further adjustment for depression, but there were still significantly higher odds of suicidal ideation for both older adults with experiences of single-type mistreatment (OR = 2.48, 95% CI = [1.97, 3.12]) and multi-type mistreatment (OR = 3.19, 95% CI = [2.25, 4.51]), as compared to those without any mistreatment. The following covariates were significantly associated with suicidal ideation: age, marital status, self-rated health, social network, frequency of contact, and depression. Particularly, depression had the highest odds for suicidal ideation (OR = 4.59, 95% CI = [3.80, 5.55]). The Hosmer–Lemeshow goodness-of-fit test showed a good fit for the model (*χ*^2^ = 5.41, df = 8, *p* = 0.71).

As a sensitivity analysis, we repeated our multivariate logistic regression using unweighted data. The results of the sensitivity analysis yielded similar results (Appendix A). Single- and multi-type mistreatment experiences were found to be associated with increased risks of depression (OR = 1.98, 95% CI = [1.66, 2.37] and OR = 3.28, 95% CI = [2.33, 4.60], respectively) and suicidal ideation (OR = 2.84, 95% CI = [2.26, 3.56] and OR = 3.38, 95% CI = [2.35, 4.87], respectively), even after adjusting for all confounding factors. The Hosmer–Lemeshow goodness-of-fit test was not significant for both models, indicating the models fit well (*χ*^2^ = 4.77, df = 8, *p* = 0.78 for depression; *χ*^2^ = 5.21, df = 8, *p* = 0.74 for suicidal ideation). We also performed an additional sensitivity analysis after excluding older adults with cognitive impairment and found similar results.

## 4. Discussion

Using a nationally representative sample of community-dwelling Korean older adults, the current study examined the associations of elder mistreatment with depression and suicidal ideation. The estimated prevalence rate of elder mistreatment was 9.9% (9.8% in the sensitivity analysis using unweighted data). The rate was similar to that observed in the same study conducted in 2014 [25]. Among the mistreated older adults, 79.4% experienced single-type mistreatment, while 20.6% suffered from multi-type (two or more types) mistreatment. Consistent with a meta-analysis based on 52 studies conducted in various countries including the US, European, and Southeast Asian regions [1], psychological mistreatment was the most common type observed in the Korean older population.

Findings from this study are consistent with our hypothesis that experiencing elder mistreatment would be associated with a higher risk of depression. Multivariate logistic regression analysis revealed that both single-type and multi-type mistreatment experiences were significantly associated with an increased risk of depression, even after adjusting for multiple potential confounding factors including socio-demographic, health-related, and social relation characteristics. The current findings are generally consistent with previous studies demonstrating that older adults with mistreatment experiences are had a higher risk of depression [12,21,26,27,28,29]. A study by Fisher and Regan [12] showed that US women aged 55 and older who experienced psychological mistreatment had significantly increased odds of having depression or anxiety. In a recent study by Chao et al. [27], US Chinese older adults with any mistreatment experiences had increased odds of clinical depression; there were statistically significant associations observed between mistreatment subtypes (psychological mistreatment, financial exploitation, and caregiver neglect) and odds of depression, but no significant association was found with either physical or sexual mistreatment. Our findings also support a study conducted in South Korea, which found that exposure to any mistreatment increases the risk of depression among both older men and women [21].

Consistent with the hypotheses, our findings provide evidence for a significant association between elder mistreatment and suicidal ideation, after adjusting for confounding factors. The current body of empirical research investigating the association between elder mistreatment and suicidal ideation is relatively limited, but some evidence does exist. In a longitudinal study conducted with 3000 Chinese older adults in the US [16], elder mistreatment was associated with a more than 2-fold increased risk of both 2-week and 12-month suicidal ideation after controlling for potential confounders (e.g., age, gender, education, income, comorbidities, depressive symptoms, and social support). A Swedish nationwide study found that psychological mistreatment in both men and women aged 65-84 was associated with a significant increase in suicidal thoughts [18]. Furthermore, exposure to physical mistreatment increased the risk for both suicidal thoughts and attempted suicide in older men [18]. In addition, several studies examining factors affecting suicidal ideation among older adults have found evidence of a strong association between elder mistreatment and suicidal ideation [17,30,31]. A study conducted by Lee and Atteraya [30] in South Korea showed that elder mistreatment experience had the highest odds ratio among the predictive factors of suicidal ideation. The results of the current study confirmed and extended previous findings by suggesting that both single- and multi-type elder mistreatment experiences are associated with increased risk for suicidal ideation in later life.

Our findings also indicated that exposure to multi-type mistreatment is greatly associated with depression and suicidal ideation. A substantial number of mistreated older individuals suffer from multi-type mistreatment in this population, and they appear to be at a much higher risk of developing depression and having suicidal ideation. However, to date, most studies have typically focused on the effects of exposure on single-type elder mistreatment, and multi-type mistreatment has received little attention in the literature. The few available studies found that multi-type mistreatment has a much higher risk of adverse physical and mental health consequences [12,20]. In a study by Simmons and Swahnberg [20], experiencing multi-type elder mistreatment was associated with adverse physical and mental health, while single-type mistreatment was not. In another study, either single- (psychological mistreatment alone) or multi-type mistreatment (combination of psychological mistreatment with other types of mistreatment) was associated with a greater risk of depression or anxiety; however, the odds were greater for multi-type (OR = 1.85, *p* < 0.01) than single-type (OR = 1.74, *p* < 0.001) [12]. Our findings suggested that, along with previous findings, the experience of multi-type mistreatment might be strongly associated with a higher risk of mental health problems in older adults.

### 4.1. Study Limitations

This study has some limitations worth noting. Firstly, because of using cross-sectional data, causal inferences cannot be addressed. For example, the data lack information on the pre-existence of depressive symptoms; we cannot determine whether the presence of depression was pre-existent or induced by mistreatment experiences. In addition, there is no information on the exact timing of suicidal ideation. This suggests that future studies should further explore the causal relationship between mistreatment, depression, and suicidal ideation. Secondly, study assessments including mistreatment and depression were based on self-reports and each subtype of mistreatment was assessed by a one-item question. Additionally, the lack of detailed information on mistreatment (e.g., severity, frequency of mistreatment) may have limited the interpretation of the results. Lastly, most of the mistreated older adults in this study had experienced psychological mistreatment. Due to the relatively low prevalence of other types of mistreatment, including physical and financial mistreatment, subgroup analyses by each mistreatment subtype were not performed. Since some studies have shown that the association between elder mistreatment and mental health outcomes may differ depending on the subtype of mistreatment, it is necessary to investigate the differences between subtypes.

### 4.2. Implications

The findings of this study have important implications for practice. Our findings suggest that single- and multi-type mistreatment experiences in old age are associated with adverse mental health outcomes, such as increasing the risk of depression and suicidal ideation. Moreover, older adults who experience multi-type mistreatment are at greater risk of developing mental health problems. For these mistreated individuals, it is important to assess their psychological status, especially depression and suicidal ideation. The findings support the need for a structured screening protocol for psychological problems in mistreated older adults. They also imply the need for tailored interventions and appropriate referral protocols. Practitioners and health care professionals who interact with mistreated older adults should play an important role in sensitively screening and caring for mistreated older adults.

There are also implications for research. Future research is needed to provide a more extensive understanding of the effects of single- and multi-type mistreatment experiences on physical and psychological health outcomes in older adults. Further research should consider whether these findings can be replicated in other samples. In addition, future studies would benefit from considering the severity and subtypes of mistreatment.

## 5. Conclusions

Elder mistreatment is prevalent in older adults in South Korea, and experiencing mistreatment can seriously affect the mental health of older adults. The findings of this nationally representative sample of older adults provide evidence for the significant associations of single- and multi-type mistreatment with depression and suicidal ideation. Additionally, the findings also provide evidence that experiencing multi-type elder mistreatment may be associated with a greater increased risk of depression and suicidal ideation. Our findings add to existing knowledge by examining the association between multi-type mistreatment and mental health outcomes.

## Figures and Tables

**Table 1 ijerph-19-11165-t001:** Characteristics of the study participants (weighted).

Variables	Total (*n* = 10,055)	Single Type of EM (*n* = 786)	Multiple Types of EM (*n* = 204)	Depressed Group (*n* = 2116)	Suicidal Ideation (*n* = 672)
	*n* (%)				
Age (years), *M* (*SD*)	73.87 (6.54)	74.40 (6.56)	74.14 (6.41)	75.81 (6.88)	73.17 (6.12)
Gender					
Female	5778 (57.5)	467 (59.5)	114 (55.9)	1381 (65.3)	411 (61.2)
Male	4277 (42.5)	319 (40.5)	90 (44.1)	735 (34.7)	261 (38.8)
Education					
<high school	7541 (75.0)	596 (75.9)	170 (83.3)	1800 (85.0)	528 (78.6)
≥high school	2514 (25.0)	190 (24.1)	34 (16.7)	316 (15.0)	144 (21.4)
Marital status					
Married	6404 (63.7)	436 (55.5)	68 (33.3)	1054 (49.8)	325 (48.4)
Unmarried	3651 (36.3)	350 (44.5)	136 (66.7)	1062 (50.2)	346 (51.6)
Living with children					
Yes	2403 (23.9)	183 (23.3)	29 (14.2)	518 (24.5)	142 (21.2)
No	7652 (76.1)	603 (76.7)	175 (85.8)	1598 (75.5)	529 (78.8)
Area					
Rural	3142 (31.3)	278 (35.3)	48 (23.3)	618 (29.2)	191 (28.4)
Urban	6913 (68.7)	508 (64.7)	157 (76.7)	1498 (70.8)	481 (71.6)
Household income ^a^					
Quintile 1	2033 (20.2)	198 (25.1)	93 (45.4)	711 (33.6)	221 (32.9)
Quintile 2	2004 (19.9)	148 (18.9)	47 (22.9)	488 (23.0)	140 (20.9)
Quintile 3	2019 (20.1)	168 (21.4)	30 (14.8)	374 (17.7)	121 (17.9)
Quintile 4	2001 (19.9)	119 (15.1)	17 (8.5)	312 (14.7)	102 (15.2)
Quintile 5	1998 (19.9)	153 (19.5)	17 (8.4)	232 (11.0)	88 (13.1)
Chronic disease					
Yes	8999 (89.5)	737 (93.7)	198 (97.0)	2042 (96.5)	644 (95.9)
No	1056 (10.5)	49 (6.3)	6 (3.0)	74 (3.5)	28 (4.1)
ADL/IADL disability ^b^					
Yes	2371 (23.6)	226 (28.7)	77 (37.7)	965 (45.6)	224 (33.4)
No	7684 (76.4)	560 (71.3)	127 (62.3)	1152 (54.4)	447 (66.6)
Cognitive impairment					
Yes	1435 (14.3)	117 (14.9)	38 (18.5)	429 (20.3)	128 (19.1)
No	8620 (85.7)	669 (85.1)	166 (81.5)	1688 (79.7)	543 (80.9)
Self-rated health					
Poor	6334 (63.0)	534 (67.9)	168 (82.3)	1908 (90.2)	561 (83.5)
Good	3721 (37.0)	252 (32.1)	36 (17.7)	208 (9.8)	111 (16.5)
Physical inactivity					
Yes	6842 (68.0)	552 (70.3)	114 (55.9)	1120 (52.9)	430 (64.0)
No	3213 (32.0)	234 (29.7)	90 (44.1)	996 (47.1)	242 (36.0)
Drinking					
Yes	2680 (26.7)	227 (28.9)	46 (22.5)	406 (19.2)	172 (25.7)
No	7375 (73.3)	559 (71.1)	158 (77.5)	1710 (80.8)	499 (74.3)
Social participation					
Yes	6044 (60.1)	457 (58.1)	81 (39.7)	725 (34.2)	314 (46.7)
No	4011 (39.9)	329 (41.9)	123 (60.3)	1392 (65.8)	358 (53.3)
Social network ^c^, *M* (*SD*)	2.27 (2.65)	2.08 (2.68)	1.07 (1.63)	1.27 (1.86)	1.91 (2.67)
Frequency of contact ^d^, *M* (*SD*)	3.05 (1.07)	2.94 (1.10)	2.36 (1.16)	2.52 (1.15)	2.73 (1.23)

Note: All estimates are weighted. ^a^ Quintile 1 = lowest income quintile, Quintile 5 = highest income quintile; ^b^ ADL = activities of daily living, IADL = instrumental ADL; ^c^ higher scores indicate a larger social network; ^d^ score 0 (never)–6 (almost every day).

**Table 2 ijerph-19-11165-t002:** Odds ratio of depression by elder mistreatment experience (weighted).

Variables	OR	95% CI	*p-*Value
Independent variables			
EM experience			
No EM	1	(ref)	
Single type of EM	1.93	[1.61, 2.32]	<0.001
Multiple types of EM	3.51	[2.52, 4.87]	<0.001
Confounding variables (covariates)			
Age (years)	0.99	[0.98, 1.00]	<0.01
Gender			
Female	1.00	[0.87, 1.15]	1.00
Male	1	(ref)	
Education			
<high school	1.18	[1.01, 1.38]	<0.05
≥high school	1	(ref)	
Marital status			
Married	0.90	[0.79, 1.03]	0.14
Unmarried	1	(ref)	
Living with children			
Yes	1.43	[1.21, 1.69]	<0.001
No	1	(ref)	
Area			
Rural	0.85	[0.75, 0.96]	<0.05
Urban	1	(ref)	
Household income			
Quintile 1	3.06	[2.42, 3.88]	<0.001
Quintile 2	2.36	[1.88, 2.95]	<0.001
Quintile 3	1.88	[1.51, 2.33]	<0.001
Quintile 4	1.51	[1.22, 1.86]	<0.001
Quintile 5	1	(ref)	
Chronic disease			
Yes	1.40	[1.06, 1.84]	<0.05
No	1	(ref)	
ADL/IADL disability			
Yes	1.76	[1.55, 2.00]	<0.001
No	1	(ref)	
Cognitive impairment			
Yes	1.11	[0.96, 1.28]	0.18
No	1	(ref)	
Self-rated health			
Poor	4.12	[3.49, 4.85]	<0.001
Good	1	(ref)	
Physical inactivity			
Yes	1.52	[1.36, 1.71]	<0.001
No	1	(ref)	
Drinking			
Yes	1.01	[0.88, 1.17]	0.85
No	1	(ref)	
Social participation			
Yes	0.59	[0.52, 0.66]	<0.001
No	1	(ref)	
Social network	0.91	[0.89, 0.94]	<0.001
Frequency of contact	0.75	[0.70, 0.79]	<0.001

Note: OR = odds ratio, CI = confidence interval.

**Table 3 ijerph-19-11165-t003:** Odds ratio of suicidal ideation by elder mistreatment experience (weighted).

Variables	Model 1			Model 2		
	OR	95% CI	*p-*Value	OR	95% CI	*p-*Value
Independent variables						
EM experience						
No EM	1	(ref)		1	(ref)	
Single type of EM	2.85	[2.28, 3.56]	<0.001	2.48	[1.97, 3.12]	<0.001
Multiple types of EM	4.21	[3.01, 5.89]	<0.001	3.19	[2.25, 4.51]	<0.001
Confounding variables (covariates)						
Age (years)	0.94	[0.93, 0.95]	<0.001	0.94	[0.93, 0.95]	<0.001
Gender						
Female	0.90	[0.73, 1.09]	0.28	0.89	[0.72, 1.09]	0.24
Male	1	(ref)		1	(ref)	
Education						
<high school	0.90	[0.73, 1.12]	0.35	0.87	[0.70, 1.08]	0.21
≥high school	1	(ref)		1	(ref)	
Marital status						
Married	0.62	[0.51, 0.76]	<0.001	0.63	[0.51, 0.77]	<0.001
Unmarried	1	(ref)		1	(ref)	
Living with children						
Yes	1.05	[0.81, 1.35]	0.72	0.96	[0.74, 1.23]	0.72
No	1	(ref)		1	(ref)	
Area						
Rural	0.93	[0.77, 1.12]	0.45	0.98	[0.81, 1.19]	0.86
Urban	1	(ref)		1	(ref)	
Household income						
Quintile 1	1.83	[1.30, 2.58]	<0.01	1.41	[0.99, 2.00]	0.05
Quintile 2	1.52	[1.09, 2.12]	<0.05	1.25	[0.89, 1.74]	0.20
Quintile 3	1.35	[0.98, 1.85]	0.07	1.17	[0.84, 1.61]	0.36
Quintile 4	1.17	[0.86, 1.60]	0.31	1.07	[0.78, 1.46]	0.69
Quintile 5	1	(ref)		1	(ref)	
Chronic disease						
Yes	1.62	[1.08, 2.44]	<0.05	1.51	[1.00, 2.30]	0.05
No	1	(ref)		1	(ref)	
ADL/IADL disability						
Yes	1.30	[1.06, 1.58]	<0.05	1.11	[0.90, 1.36]	0.33
No	1	(ref)		1	(ref)	
Cognitive impairment						
Yes	1.25	[1.00, 1.55]	<0.05	1.21	[0.97, 1.50]	0.10
No	1	(ref)		1	(ref)	
Self-rated health						
Poor	2.35	[1.87, 2.96]	<0.001	1.77	[1.39, 2.25]	<0.001
Good	1	(ref)		1	(ref)	
Physical inactivity						
Yes	1.01	[0.85, 1.21]	0.90	0.87	[0.72, 1.05]	0.14
No	1	(ref)		1	(ref)	
Drinking						
Yes	1.14	[0.93, 1.40]	0.22	1.14	[0.92, 1.40]	0.24
No	1	(ref)		1	(ref)	
Social participation						
Yes	0.78	[0.64, 0.94]	<0.01	0.92	[0.75, 1.12]	0.38
No	1	(ref)		1	(ref)	
Social network	1.03	[0.99, 1.06]	0.16	1.05	[1.01, 1.09]	<0.05
Frequency of contact	0.82	[0.75, 0.89]	<0.001	0.89	[0.82, 0.98]	<0.05
Depression						
Yes				4.59	[3.80, 5.55]	<0.001
No				1	(ref)	

Note: OR = odds ratio, CI = confidence interval.

## Data Availability

We used data from the “2017 National Survey of Living Conditions and Welfare Needs of Korean Older Persons” conducted by the Korean Ministry of Health and Welfare and the Korea Institute for Health and Social Affairs (KIHASA). Data can be accessible upon request.

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
