# Peer review of "Elder Mistreatment as a Risk Factor for Depression and Suicidal Ideation in Korean Older Adults"

_ijerph, 2022, doi:10.3390/ijerph191811165_

Round 1

Reviewer 1 Report

Interesting and relevant article about the mistreatment of the elderly population and its repercussions on depression and suicidal ideation, which are prevalent mental health problems in this population, having confirmed the hypotheses raised in the study.

I would recommend including a consideration regarding the timing of the measurement of depression and suicidal ideation, since while the first measures the present, the second refers to life.

Another recommendation is to explain why it is stated that "Since the establishment of the Elder Protective Services agency (EPS) in 2004, there has been a considerable influence of reported cases of mistreatment." and include some conclusion in this regard derived from the results of this research "using a nationally representative sample of community-dwelling Korean older adults."

Round 2

Reviewer 2 Report

Review the questions and suggestions raised.

Important issues have been found to have been improved.

Therefore, consideration could be given to accepting publication.